# Preparation and Photocatalytic Performance for Degradation of Rhodamine B of AgPt/Bi_4_Ti_3_O_12_ Composites

**DOI:** 10.3390/nano10112206

**Published:** 2020-11-05

**Authors:** Gaoqian Yuan, Gen Zhang, Kezhuo Li, Faliang Li, Yunbo Cao, Jiangfeng He, Zhong Huang, Quanli Jia, Shaowei Zhang, Haijun Zhang

**Affiliations:** 1The State Key Laboratory of Refractories and Metallurgy, Wuhan University of Science and Technology, Wuhan 430081, China; yuangaoqian@126.com (G.Y.); zhang0812m@163.com (G.Z.); likezhuo0206@126.com (K.L.); 15623242376@163.com (Y.C.); hejiangfeng823102@163.com (J.H.); huangzhong@wust.edu.cn (Z.H.); 2Jiangxi Engineering Research Center of Industrial Ceramics, Pingxiang 337022, China; 3Henan Key Laboratory of High Temperature Functional Ceramics, Zhengzhou University, 75 Daxue Road, Zhengzhou 450052, China; jiaquanli@zzu.edu.cn; 4College of Engineering, Mathematics and Physical Sciences, University of Exeter, Exeter EX4 4QF, UK; s.zhang@exeter.ac.uk

**Keywords:** photocatalytic, Rhodamine B, photoreduction, Bi_4_Ti_3_O_12_, AgPt/Bi_4_Ti_3_O_12_

## Abstract

Loading a noble metal on Bi_4_Ti_3_O_12_ could enable the formation of the Schottky barrier at the interface between the former and the latter, which causes electrons to be trapped and inhibits the recombination of photoelectrons and photoholes. In this paper, AgPt/Bi_4_Ti_3_O_12_ composite photocatalysts were prepared using the photoreduction method, and the effects of the type and content of noble metal on the photocatalytic performance of the catalysts were investigated. The photocatalytic degradation of rhodamine B (RhB) showed that the loading of AgPt bimetallic nanoparticles significantly improved the catalytic performance of Bi_4_Ti_3_O_12_. When 0.10 wt% noble metal was loaded, the degradation rate for RhB of Ag_0.7_Pt_0.3_/Bi_4_Ti_3_O_12_ was 0.027 min^−1^, which was respectively about 2, 1.7 and 3.7 times as that of Ag/Bi_4_Ti_3_O_12_, Pt/Bi_3_Ti_4_O_12_ and Bi_4_Ti_3_O_12._ The reasons may be attributed as follows: (i) the utilization of visible light was enhanced due to the surface plasmon resonance effect of Ag and Pt in the visible region; (ii) Ag nanoparticles mainly acted as electron acceptors to restrain the recombination of photogenerated electron-hole pairs under visible light irradiation; and (iii) Pt nanoparticles acted as electron cocatalysts to further suppress the recombination of photogenerated electron-hole pairs. The photocatalytic performance of Ag_0.7_Pt_0.3_/Bi_4_Ti_3_O_12_ was superior to that of Ag/Bi_4_Ti_3_O_12_ and Pt/Bi_3_Ti_4_O_12_ owing to the synergistic effect between Ag and Pt nanoparticles.

## 1. Introduction

The rapid growth of the chemical industry has led to a large volume of organic dye wastewater. Most of the organic pollutants have carcinogenic effects, posing a huge threat to organisms and human health [1,2,3]. Therefore, organic pollutants must be detoxified before they enter aquatic ecosystems. Some traditional wastewater treatment processes, such as physical adsorption, chemical oxidation and microbial treatment [4,5,6], have been proposed to deal with organic dyestuff sewage. However, these methods have suffered from several shortcomings including low efficiency, secondary pollution, and mild degree of harmlessness of pollutant [7,8,9]. Photocatalysis technology, which can non-selectively oxidize and degrade all kinds of organic matter with the ability of deep oxidation, is considered as an acceptable process for the treatment of organic contaminants since only water and non-toxic inorganic substances are produced. In the process of photocatalysis, photocatalysts are always required for achieving excellent catalytic effect. However, for the most widely used semiconductor photocatalysts (such as TiO_2_ [10], SrTiO_3_ [11], ZnS [12] and ZnO [13]), the utilization of solar energy is usually very low since their bandgap width *E*_g_ is always larger than 3.0 eV and can only just absorb ultraviolet light (only 4% of the total sunlight). Thus, how to improve the usage of solar energy in the photocatalytic removal of organic pollutants has become an important issue.

Bi_4_Ti_3_O_12_ can respond to visible light since its bandgap is about 2.9 eV, and shows strong photocatalytic activity for the removal of organic pollutants [14,15,16]. However, the usage of Bi_4_Ti_3_O_12_ is still limited because the photon efficiency is low and the photogenerated electrons and holes are easy to recombine. Noble metal nanoparticles (NPs) can be used as the accumulation ground of photogenerated electrons to facilitate the catalytic reactions involving electrons, and thus are widely employed as modifiers to enhance the photocatalytic performance of semiconductors [17,18,19]. When precious metals and semiconductors contact together, electrons on the surface of the latter will migrate to the surface of the former until their Fermi energy levels equalize. Since charges on the metal surface and holes on the semiconductor surface are both excessive, a Schottky barrier can be formed on the metal-semiconductor interface. As a result, the separation of photogenerated electrons and holes is promoted, and an improved photocatalytic performance is obtained [20,21,22]. 

In the present paper, Bi_4_Ti_3_O_12_ nanosheets were firstly prepared by a molten salt method, and then AgPt bimetallic NPs were assembled on the prepared Bi_4_Ti_3_O_12_ nanosheets via an in situ photoreduction method to prepare AgPt/Bi_4_Ti_3_O_12_ composites, finally the photocatalytic performance of as-prepared AgPt/Bi_4_Ti_3_O_12_ composites on the degradation of rhodamine B (RhB) under visible light was investigated. As far as we know, there has been no study on the decoration of AgPt bimetallic NPs on Bi_4_Ti_3_O_12_ photocatalyst. 

## 2. Materials and Methods 

### 2.1. Materials

Titanium oxide (TiO_2_) and bismuth oxide (Bi_2_O_3_) were purchased from Shanghai Makclin Biochemical Co., Ltd. (Shanghai, China). Silver nitrate (AgNO_3_) was purchased from Tianjin Kaitong Chemical Reagent Co., Ltd. (Tianjin, China). Chloroplatinic acid hexahydrate (H_2_PtCl_6_·6H_2_O) was purchased from Shanghai Aladdin Bio-chem Technology Co., Ltd. (Shanghai, China). Potassium chloride (KCl) and sodium chloride (NaCl) were purchased from Tianjin Bodi Chemical Reagent Co., Ltd. (Tianjin, China) All chemicals were used as purchased without further purification. 

### 2.2. Preparation of Bi_4_Ti_3_O_12_ Powders 

Preparation of Bi_4_Ti_3_O_12_ powders via the molten salt method was similar to our previously published paper [23,24,25,26,27]. Typically, stoichiometric amounts of Bi_2_O_3_, TiO_2_, NaCl and KCl were weighed firstly according to a predetermined ratio shown in Table 1. After that, the raw materials and molten salt medium were mixed in a planetary ball mill for 3 h under a rotating speed of 300 r/min with ethanol as milling medium. Subsequently, the mixed powders were dried and then subjected to 2 h heating treatment at 700, 800 and 900 °C in a muffle furnace. Finally, Bi_4_Ti_3_O_12_ powders were obtained after washing, filtration and drying, and the samples were labeled as Bi_4_Ti_3_O_12_-T-M (T is the heat treatment temperature, M is the mass ratio of molten salt medium and raw material). 

### 2.3. Assembly of Ag and Pt Nanoparticles (NPs) on Bi_4_Ti_3_O_12_ Nanosheets 

Ag/Bi_4_Ti_3_O_12_, Pt/Bi_4_Ti_3_O_12_ and AgPt/Bi_4_Ti_3_O_12_ were prepared by a photoreduction method. The whole preparation procedure involved four steps as follows: (i) 1 g as-prepared Bi_4_Ti_3_O_12_ powders were dispersed in 200 mL deionized water; (ii) 0.1 mol/L AgNO_3_ solution, 7.72 mmol/L H_2_PtCl_6_·6H_2_O solution and the mixed solution of AgNO_3_ and H_2_PtCl_6_·6H_2_O was separately added to the as-prepared Bi_4_Ti_3_O_12_ suspension with the predetermined proportions shown in Table 2; (iii) The as-prepared suspensions were irradiated under a 300 W xenon lamp for 60 min (a 400 nm filter was used to block ultraviolet (UV) light (*λ* < 400 nm)). In this process, Ag^+^ and Pt^4+^ was respectively reduced to Ag NPs and Pt NPs; and iv) The Ag/Bi_4_Ti_3_O_12_, Pt/Bi_4_Ti_3_O_12_ and AgPt/Bi_4_Ti_3_O_12_ powders were obtained after the mixture was filtered, washed and dried (drying conditions: 80 °C for 12 h in an electric drying oven).

### 2.4. Characterization

X-ray diffraction (XRD) was performed on MiniFlex 600 with Cu Kα radiation (*λ* = 1.54178 Å) to investigate the crystal structure of the as-prepared powders. The field-emission scanning electron microscopy (FE-SEM) images, transmission electron microscopy (TEM) images, selected area electron diffraction (SAED) images and energy dispersive spectroscopy (EDS) elemental mapping images were respectively taken on a JEOL JSM-6700F SEM and JEM-2100 HR TEM to observe the microstructure of the as-prepared catalysts. Fourier transform infrared (FT-IR) spectra ranged from 3000–450 cm^−1^ were recorded on a Nicolet iS50 spectrometer in air at room temperature to differentiate the functional groups formed on the surface of as-prepared catalysts. Chemical composition of as-prepared catalysts was analyzed by the inductively coupled plasma mass spectroscopy (ICP-MS, Spectro Flame, Spectro Analytical Instrument, Kleve, Germany). X-ray photoelectron spectroscopy (XPS) measurements were performed using an AMICUS ESCA 3400 XPS with Al Kα radiation. In order to investigate the elemental information (Bi, Ti, O, Ag and Pt), all XPS spectra were calibrated by shifting the detected adventitious carbon C 1s peak to 284.8 eV. A RF-6000 fluorescence spectrophotometer was used to measure the photoluminescence (PL) spectra of the samples (excitation wavelength: 320 nm). Ultraviolet-visible (UV-Vis) absorption spectra were recorded using a UV-Vis spectrophotometer (Shimadzu UV-3600, Kyoto, Japan) to study the responsive behavior under UV and visible light irradiation of the as-prepared catalysts.

### 2.5. Photocatalytic Activity

RhB aqueous solution was chosen to simulate wastewater and the variation of its concentration under the irradiation of xenon lamp (*λ* > 400 nm) was measured at *λ* = 554 nm by a UV-Vis spectrophotometer. Four steps were performed to investigate the performance of the as-prepared catalysts: (i) 3.2 mg/L RhB solution was prepared firstly and then its absorption spectrum was determined by a UV-Vis spectrophotometer; (ii) 200 mg as-prepared photocatalysts were put into 200 mL RhB solution with an initial concentration of 3.2 mg/L, and then, in order to achieve adsorption-desorption equilibrium, the obtained mixture was stirred in the dark for 1 h. After that, 3 mL solution was taken out to be centrifuged and the supernatant was taken out for absorption spectrum measurement, from which the adsorbed amount of RhB can be calculated; (iii) The residual mixture was exposed to a xenon lamp irradiation (*λ* > 400 nm) for 30 min and then 3 mL solution was taken out to be centrifuged, and the supernatant was taken out to measure its absorption spectrum; and (iv) Step 3 was repeated until the identified absorption spectra were almost unchanged. In order to eliminate the influence of temperature on the photocatalytic behavior, the whole experimental process was carried out in a cooling cycle device. 

### 2.6. Detection of Reactive Species

In a typical reactive species trapping experiment, three reactions were carried out to distinguish the reactive species of hole (h^+^), hydroxy radical (•OH) and superoxide radical (•O^2−^) for the photocatalytic degradation process. At first, 200 mL RhB solution and 0.2 g as-prepared catalyst were respectively mixed with 20 mL ethanol (EtOH), 0.2 mmol benzoquinone (BQ) and 2 mmol triethanolamine (TEOA). Then the mixtures were stirred for 1 h in dark for achieving adsorption-desorption equilibrium. Finally, the catalytic process as described in Section 2.5 was carried out. 

## 3. Results

Figure 1 showed the XRD patterns of Bi_4_Ti_3_O_12_ powders (Bi_4_Ti_3_O_12_-800-1) prepared by molten salt method and (AgPt)_0.001_/Bi_4_Ti_3_O_12_ composites prepared by a photoreduction method. It can be clearly seen that the diffraction peaks of the as-prepared Bi_4_Ti_3_O_12_ matched well with the standard card JCPDS-01-080-2143, indicating the crystallinity of synthesized Bi_4_Ti_3_O_12_ was high. The diffraction peaks of Ag and Pt NPs were not observed, and the reason may be ascribed to the low amount of loaded AgPt bimetallic NPs (0.1 wt%).

Figure 2 presents the SEM images of as-prepared Bi_4_Ti_3_O_12_ and (AgPt)_0.001_/Bi_4_Ti_3_O_12_ composite photocatalysts prepared by the photoreduction method. It revealed that all the samples showed a lamellar structure with a grain size of 1–5 μm, and that Ag and Pt NPs were too small to be observed in the SEM images.

The representative TEM image of the as-prepared (Ag_0.7_Pt_0.3_)_0.001_/Bi_4_Ti_3_O_12_ photocatalyst is shown in Figure 3a, confirming that (Ag_0.7_Pt_0.3_)_0.001_/Bi_4_Ti_3_O_12_ photocatalyst exhibited a nano-sheet structure with proper thickness. As can be seen from the SAED pattern (Figure 3b), single crystalline Bi_4_Ti_3_O_12_ was prepared. Figure 3c obviously showed that Ag and Pt NPs were modified on the surface of the (Ag_0.7_Pt_0.3_)_0.001_/Bi_4_Ti_3_O_12_ sample (dashed area). Furthermore, two different d-spacing values of 0.226 nm and 0.239 nm were calculated from the lattice fringes of loaded particles (Figure 3d), which respectively correspond to the (111) crystal plane of cubic Pt and the (111) crystal plane of metallic Ag [28,29]. Figure 4, Figure 5 and Figure 6 respectively presented the TEM images and the corresponding EDS elemental mapping images of (Ag_0.7_Pt_0.3_)_0.001_/Bi_4_Ti_3_O_12_, Ag_0.001_/Bi_4_Ti_3_O_12_, and Pt_0.001_/Bi_4_Ti_3_O_12_ photocatalysts prepared by photoreduction method. It can be clearly seen from Figure 4a that a large amount of spherical particles with an average size of 9 nm were randomly distributed over the surface of as-prepared Bi_4_Ti_3_O_12_. EDS mapping results presented in Figure 4b–f indicated that the spherical particles were Ag and Pt NPs, confirming (Ag_0.7_Pt_0.3_)_0.001_/Bi_4_Ti_3_O_12_ composite photocatalysts were successfully prepared. Similarly, Figure 5 and Figure 6 separately proved that Ag_0.001_/Bi_4_Ti_3_O_12_ and Pt_0.001_/Bi_4_Ti_3_O_12_ were successfully synthesized. 

ICP-MS results presented in Table 3 revealed that the content of Ag in Ag_0.001_/Bi_4_Ti_3_O_12_ and Pt in Pt_0.001_/Bi_4_Ti_3_O_12_ photocatalysts was respectively 0.0347 wt% and 0.0782 wt%, while the content of Ag and Pt in (Ag_0.7_Pt_0.3_)_0.001_/Bi_4_Ti_3_O_12_ was respectively 0.0122 wt% and 0.0144 wt%, demonstrating that Ag and Pt have been successfully loaded on Bi_4_Ti_3_O_12_ nanosheets. However, the actual amount was lower than that theoretical value, indicating that the Ag^+^ and Pt^4+^ were not completely photo-reduced and loaded on Bi_4_Ti_3_O_12_ under present irradiation of Xe lamp.

XPS was performed to investigate the elemental information of as-prepared (Ag_0.7_Pt_0.3_)_0.001_/Bi_4_Ti_3_O_12_ photocatalyst. As shown in Figure 7a, Bi, Ti, O, Ag and Pt elements were identified, demonstrating the successful preparation of the (Ag_0.7_Pt_0.3_)_0.001_/Bi_4_Ti_3_O_12_ composite photocatalyst. The high-resolution XPS spectrum of Bi-4f, Ti-2p, O-1s, Ag-3d and Pt-4f were investigated (Figure 7b–f). Two peaks at 159.2 and 164.5 eV were respectively ascribed to Bi 4f_7/2_ and Bi 4f_5/2_ (Figure 7b), indicating that only Bi^3+^ existed in the (Ag_0.7_Pt_0.3_)_0.001_/Bi_4_Ti_3_O_12_ composite [30,31,32]. The high resolution XPS spectrum of Ti 2p can be deconvoluted into three peaks at 458.1 eV, 463.8 eV and 466.2 eV (Figure 7c), which were assigned to Ti 2P_3/2_, Ti 2p_1/2_ and Bi 4d_3/2_, respectively [30,31,32], demonstrating that only Ti^4+^ specie existed in the (Ag_0.7_Pt_0.3_)_0.001_/Bi_4_Ti_3_O_12_ composite. From the high-resolution XPS spectrum of O 1s (Figure 7d), the peak at 529.9 eV was assigned to lattice oxygen in Bi_4_Ti_3_O_12_ and the peak at 532.2 eV was ascribed to the oxygen adsorbed on the surface of as-prepared (Ag_0.7_Pt_0.3_)_0.001_/Bi_4_Ti_3_O_12_ [30]. As shown in Figure 7e, peaks at 368.1 and 374.1 eV were, respectively, assigned to Ag 3d_5/2_ and Ag 3d_3/2_, providing conclusive evidence for Ag metal in the as-prepared (Ag_0.7_Pt_0.3_)_0.001_/Bi_4_Ti_3_O_12_ photocatalyst. As presented in Figure 7f, an asymmetric peak at 74.35 eV assigned to Pt 4f_5/2_ was observed, illustrating the existence of Pt metal in the as-prepared (Ag_0.7_Pt_0.3_)_0.001_/Bi_4_Ti_3_O_12_ sample [33,34]. Since no additional peaks were observed in the Ag 3d and Pt 4f spectra, it can be determined that Ag and Pt do not exist in the form of oxidation states.

Fourier transform infrared (FTIR) spectra further offered the functional group information of the Bi_4_Ti_3_O_12_ and (Ag_0.7_Pt_0.3_)_0.001_/Bi_4_Ti_3_O_12_ (Figure 8). The peak at 832 cm^−1^ belonged to the Bi-O characteristic stretching vibration, the strong peak at 573 cm^−1^ and the weak peak at 472 cm^−1^ belonged to the stretching vibration of Ti-O [35,36]. These results indicated that the structure of orthorhombic phase Bi_4_Ti_3_O_12_ was not damaged by loading of Ag and Pt NPs. Besides, no characteristic peaks of Pt and Ag oxides were observed, indicating that both Pt and Ag existed in metallic state in (Ag_0.7_Pt_0.3_)_0.001_/Bi_4_Ti_3_O_12_, which was in line with the XPS results presented the Figure 7. 

Figure 9 showed the UV-Vis absorption spectra of pristine Bi_4_Ti_3_O_12_ and as-prepared (AgPt)_0.001_/Bi_4_Ti_3_O_12_ composites. Obviously, (AgPt)_0.001_/Bi_4_Ti_3_O_12_ composites exhibited stronger visible light absorption than pristine Bi_4_Ti_3_O_12_. As reported in previous works, the enhancement in visible light absorption may be dominated by the surface plasma resonance effect of Ag and Pt NPs [37,38]. When the internal electron oscillation frequency of the AgPt bimetallic NPs was equal to the frequency of the irradiated light, local surface plasmon resonance was induced and then the visible light absorption of (AgPt)_0.001_/Bi_4_Ti_3_O_12_ was improved. Meanwhile, no obvious change was observed in the absorption edge for all (AgPt)_0.001_/Bi_4_Ti_3_O_12_ samples, suggesting that the bandgap of Bi_4_Ti_3_O_12_ and (AgPt)_0.001_/Bi_4_Ti_3_O_12_ photocatalysts was almost the same.

Figure 10a showed the dependence of C/C_0_ (C_0_ and C are respectively the initial and instantaneous concentration of the RhB solution) on irradiation time during the RhB degradation process when Bi_4_Ti_3_O_12_ and (AgPt)_0.001_/Bi_4_Ti_3_O_12_ were used as photocatalysts. It can be seen that the degradation rate ((C_0_ − C)/C_0_ × 100%) of RhB increased from about 69.7% to about 91.5% after irradiation of 90 min when 0.1 wt% Ag_0.7_Pt_0.3_ bimetallic NPs were loaded on Bi_4_Ti_3_O_12_. Additionally, the photocatalytic activity of (AgPt)_0.001_/Bi_4_Ti_3_O_12_ was monitored by varying the mass ratio of Ag to Pt, and the (Ag_0.7_Pt_0.3_)_0.001_/Bi_4_Ti_3_O_12_ catalysts displayed the best photocatalytic activity for RhB degradation.

In order to investigate the reaction kinetics of Bi_4_Ti_3_O_12_ and (AgPt)_0.001_/Bi_4_Ti_3_O_12_ photocatalysts for RhB degradation, relationship between Ln(C_0_/C) and irradiation time was measured and the results were shown in Figure 10b. All the photocatalytic reactions can be fitted well by pseudo-first-order kinetics,
Ln (C/C_0_) = −kt(1)

When Ln(C/C_0_) was plotted versus irradiation time t, the apparent reaction rate k can be obtained by calculating the slope of the fitted curves. The results shown in Figure 10b demonstrate that the k value relied heavily on the loading of AgPt bimetallic particles, and that the highest k value of RhB degradation catalyzed by (Ag_0.7_Pt_0.3_)_0.001_/Bi_4_Ti_3_O_12_ was calculated as 0.027 min^−1^, which was 3.7 times of that by Bi_4_Ti_3_O_12_. 

The photocatalytic activities of Ag_0.001_/Bi_3_Ti_4_O_12_ and Pt_0.001_/Bi_3_Ti_4_O_12_ composite photocatalysts prepared by the photocatalytic reduction method on the degradation of RhB after a 150 min illumination were also investigated (Figure 11). Figure 11a revealed that the degradation rate of RhB catalyzed by Pt_0.001_/Bi_4_Ti_3_O_12_ reached as high as 92.4%, which was higher than that by Ag_0.001_/Bi_4_Ti_3_O_12_ (87.7%) and by Bi_4_Ti_3_O_12_ (69.3%). Meanwhile, the apparent reaction rate k in the Pt_0.001_/Bi_4_Ti_3_O_12_ and Ag_0.001_/Bi_4_Ti_3_O_12_ catalyzed system was separately calculated as 0.01561 min^−1^ and 0.01366 min^−1^ which was, respectively, 2.1 and 1.8 times that in Bi_4_Ti_3_O_12_ case (Figure 11b). And excitingly, (AgPt)_0.001_/Bi_4_Ti_3_O_12_ composites displayed higher photocatalytic activity towards RhB degradation than Ag_0.001_/Bi_4_Ti_3_O_12_ and Pt_0.001_/Bi_4_Ti_3_O_12_. The reasons may be as follows: (i) the surface plasma resonance effect of Ag and Pt NPs enhanced the visible light absorption (Figure 8) [34]; (ii) Ag NPs can be used as electronic acceptors with excellent electrical conductivity to promote the separation of electron-hole pairs; and (iii) Pt NPs can be applied as electron cocatalyst to accelerate the capture of photogenerated electron, and then facilitated the proton reduction reaction [39,40].

Hole and radical trapping experiments were carried out to investigate the effective reactants in the photocatalytic RhB degradation process. TEOA, EtOH and BQ were respectively added to RhB solution as capture agents for h^+^, •OH, and •O^2−^ scavengers [41,42]. The main active species were determined by the change of degradation effect after the photocatalysis experiment. As can be seen from Figure 12a, the photocatalytic performance of RhB removal was almost unchanged by introducing EtOH, implying that •OH played a tiny effect on the degradation of RhB. By contrast, after adding TEOA or BQ, the photocatalytic RhB degradation activity of (Ag_0.7_Pt_0.3_)_0.001_/Bi_4_Ti_3_O_12_ was remarkably suppressed and the corresponding efficiencies for photodegradation of RhB were calculated as low as 9.4% or 10.8%, respectively (Figure 12b). It can thus be reasonably concluded that h^+^ and •O^2−^ were the active groups in the present (Ag_0.7_Pt_0.3_)_0.001_/Bi_4_Ti_3_O_12_ photocatalyzed process.

In order to study the separation behavior of electron hole pairs, photoluminescence (PL) spectroscopy was implemented. As shown in Figure 12c, phase pure Bi_4_Ti_3_O_12_, Pt_0.001_/Bi_4_Ti_3_O_12_, Ag_0.001_/Bi_4_Ti_3_O_12_ and (Ag_0.7_Pt_0.3_)_0.001_/Bi_4_Ti_3_O_12_ samples exhibit a similar PL emission curve when they were excited by a 320 nm light. However, even though the characteristic peaks of the four PL curves were all observed at about 400 nm, the PL spectroscopy of (Ag_0.7_Pt_0.3_)_0.001_/Bi_4_Ti_3_O_12_ composite exhibited the lowest emission intensity, illustrating that the separation rate of photogenerated electron hole pairs in the (Ag_0.7_Pt_0.3_)_0.001_/Bi_4_Ti_3_O_12_ sample was the highest [43,44]. As a result, more active species were formed to participate in the degradation of RhB in the case.

The stability of the photocatalyst was very important for its practical application. Therefore, the cycle experiment of RhB degradation was implemented (Figure 12d). To our amazement, the RhB degradation efficiency excited by (Ag_0.7_Pt_0.3_)_0.001_/Bi_4_Ti_3_O_12_ photocatalyst decreased slightly after three consecutive cycles. This demonstrated that the as-prepared (Ag_0.7_Pt_0.3_)_0.001_/Bi_4_Ti_3_O_12_ catalyst had good photocatalytic stability for degradation of RhB.

Based on the results obtained, a possible degradation mechanism of RhB degradation over AgPt/Bi_4_Ti_3_O_12_ was proposed (Figure 13) and described as follows: (i) under the irradiation of light, photogenerated electrons e^−^ and holes h^+^ were generated on the surface of Bi_4_Ti_3_O_12_ [45]. Generally, h^+^ can react with H_2_O or OH^−^ to form active hydroxyl •OH, e^−^ can combine with O_2_ to produce superoxide radical •O^2−^. Since h^+^, •OH and •O^2−^ all have strong oxidability, they can react directly with RhB to generate water, CO_2_ and inorganic small molecules [46,47,48,49]. However, the photogenerated h^+^ and e^−^ will recombine rapidly on the surface of Bi_4_Ti_3_O_12_ and then the photocatalytic activity was reduced; (ii) when Ag NPs were modified on the surface of Bi_4_Ti_3_O_12_, the photogenerated h^+^ and e^−^ will be redistributed since the Fermi level of Ag was lower than that of Bi_4_Ti_3_O_12_ [50]. Due to the higher Fermi level of Bi_4_Ti_3_O_12_, photogenerated e^−^ transferred from the surface of Bi_4_Ti_3_O_12_ to Ag NPs with lower Fermi level and thus Schottky barrier was formed on their interface [51]. Therefore, the photogenerated electrons e^−^ were trapped and the recombination of photogenerated electron-hole pairs was inhibited [34,38,52,53,54]. As a result, more photogenerated h^+^ was left to oxidize the RhB (Equations (2)–(10)); (iii) Pt NPs cocatalysts can provide adsorption sites for protons as electron sinks, and thus the number of h^+^ was further improved [21] and (iv) due to the surface plasma resonance effect of Ag and Pt NPs, the absorption of visible light by Bi_4_Ti_3_O_12_ was greatly improved [18,55], and more electrons and holes were generated. The reaction formulas were as follows:Bi_4_Ti_3_O_12_ + hν→ Bi_4_Ti_3_O_12_(e^−^+h^+^)(2)
Bi_4_Ti_3_O_12_(e^−^) + Ag/Pt→ Bi_4_Ti_3_O_12_ + Ag/Pt(e^−^)(3)
Bi_4_Ti_3_O_12_(h^+^) + H_2_O→ •OH+H^+^(4)
Bi_4_Ti_3_O_12_(h^+^) + OH^−^→ •OH(5)
Bi_4_Ti_3_O_12_(e^−^) + O_2_→ Bi_4_Ti_3_O_12_ + •O^2−^(6)
Ag/Pt (e^−^) +O_2_→ Ag/Pt + •O^2-^(7)
•OH + RhB → CO_2_ + H_2_O +(8)
•O^2−^ + RhB→ CO_2_ + H_2_O +‥(9)
Bi_4_Ti_3_O_12_(h^+^) + RhB → CO_2_ + H_2_O +(10)

## 4. Conclusions

Bi_4_Ti_3_O_12_ particles with lamellar structure were prepared firstly by a molten salt method as precursor, and then Ag, Pt and AgPt bimetallic NPs were loaded on the as-prepared sheet-like Bi_4_Ti_3_O_12_ to synthesize AgPt/Bi_4_Ti_3_O_12_ composites through the photoreduction procedure. The results revealed that the crystal structure and morphology of the as-prepared Bi_4_Ti_3_O_12_ were almost unchanged by loading Ag, Pt and AgPt bimetallic NPs.

It is exciting to note that the photocatalytic activity of Bi_4_Ti_3_O_12_ for RhB degradation was sharply enhanced via loading Ag, Pt or AgPt bimetallic NPs. When the loading amount was fixed at 0.1 wt%, the apparent reaction rate of the as-prepared Ag/Bi_4_Ti_3_O_12_ and Pt/Bi_4_Ti_3_O_12_ photocatalysts and Ag_0.7_Pt_0.3_/Bi_4_Ti_3_O_12_ composite photocatalysts was respectively calculated as 0.01366 min^−1^, 0.01561 min^−1^ and 0.027 min^−1^, which was respectively 1.8 times, 2.1 times and 3.7 times that of pristine Bi_4_Ti_3_O_12_ (0.00744 min^−1^). The enhanced activity may be ascribed to the surface plasma resonance effect of noble metal and Schottky barrier formed on the interface between noble metal NPs and Bi_4_Ti_3_O_12_. The reactive species trapping experiment demonstrated that h^+^ and •O^2−^ played the main role in present (Ag_0.7_Pt_0.3_)_0.001_/Bi_4_Ti_3_O_12_ photocatalyzed process, and the (Ag_0.7_Pt_0.3_)_0.001_/Bi_4_Ti_3_O_12_ catalyst exhibited good photocatalytic stability for RhB degradation.

## Figures and Tables

**Figure 1 nanomaterials-10-02206-f001:**
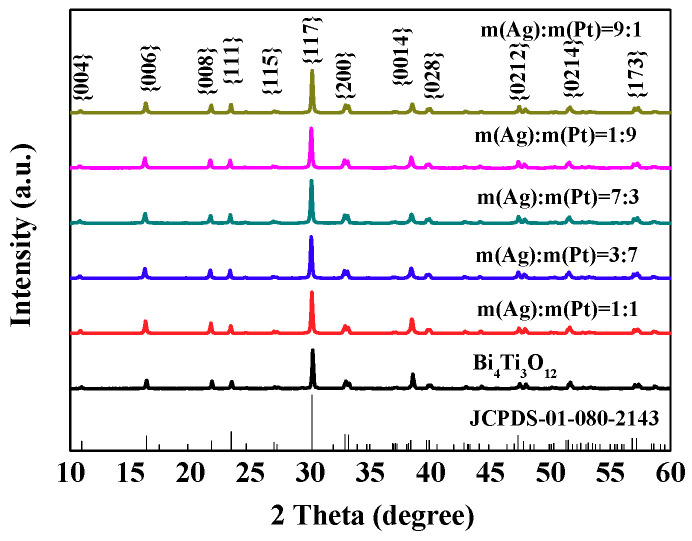
X-ray diffraction (XRD) patterns of Bi_4_Ti_3_O_12_ and (AgPt)_0.001_/Bi_4_Ti_3_O_12_ composite photocatalysts prepared by photoreduction method.

**Figure 2 nanomaterials-10-02206-f002:**
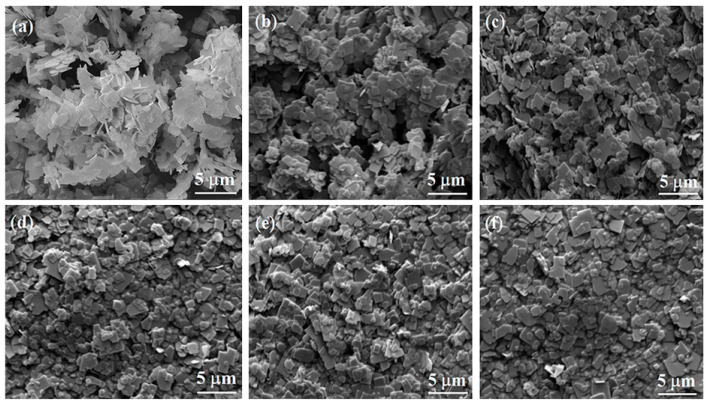
Scanning electron microscopy (SEM) images of pristine Bi_4_Ti_3_O_12_ and (AgPt)_0.001_/Bi_4_Ti_3_O_12_ composite photocatalysts prepared by photoreduction method: (**a**) Bi_4_Ti_3_O_12_; (**b**)m(Ag):m(Pt) = 1:1; (**c**) m(Ag):m(Pt) = 7:3; (**d**) m(Ag):m(Pt) = 3:7; (**e**) m(Ag):m(Pt) = 9:1 and (**f**) m(Ag):m(Pt) = 1:9.

**Figure 3 nanomaterials-10-02206-f003:**
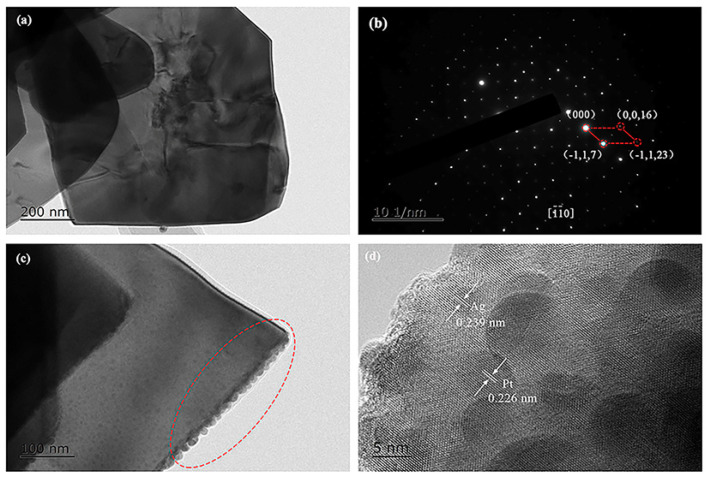
Transmission electron microscope (TEM) images (**a**,**c**), SAED image (**b**) and high-resolution TEM (HRTEM) image (**d**) of (Ag_0.7_Pt_0.3_)_0.001_/Bi_4_Ti_3_O_12_.

**Figure 4 nanomaterials-10-02206-f004:**
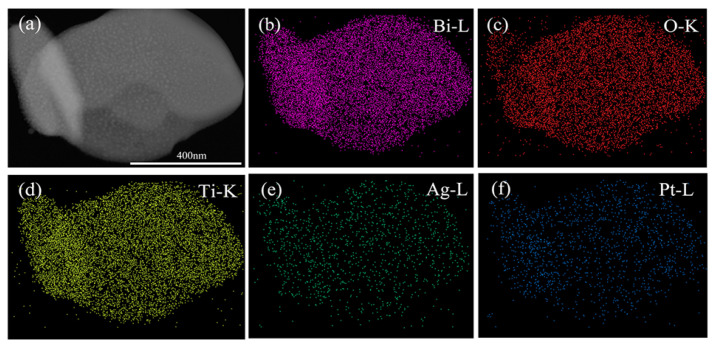
TEM image (**a**) and EDS maps of Bi, O, Ti, Ag and Pt (**b**–**f**) of as-prepared (Ag_0.7_Pt_0.3_)_0.001_/Bi_4_Ti_3_O_12_.

**Figure 5 nanomaterials-10-02206-f005:**
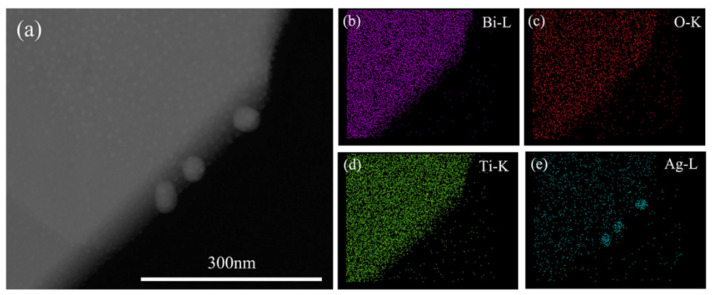
TEM image (**a**) and EDS maps of Bi, O, Ti and Ag (**b**–**e**) of as-prepared Ag_0.001_/Bi_4_Ti_3_O_12_.

**Figure 6 nanomaterials-10-02206-f006:**
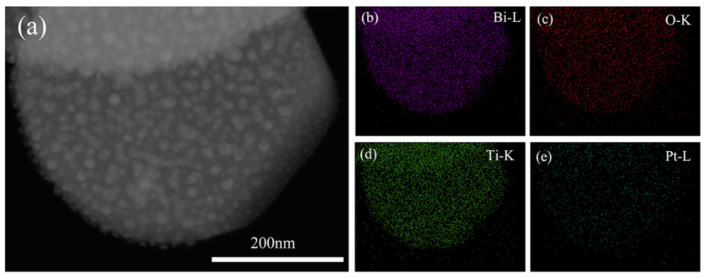
TEM image (**a**) and EDS maps of Bi, O, Ti and Pt (**b**–**e**) of as-prepared Pt_0.001_/Bi_4_Ti_3_O_12_.

**Figure 7 nanomaterials-10-02206-f007:**
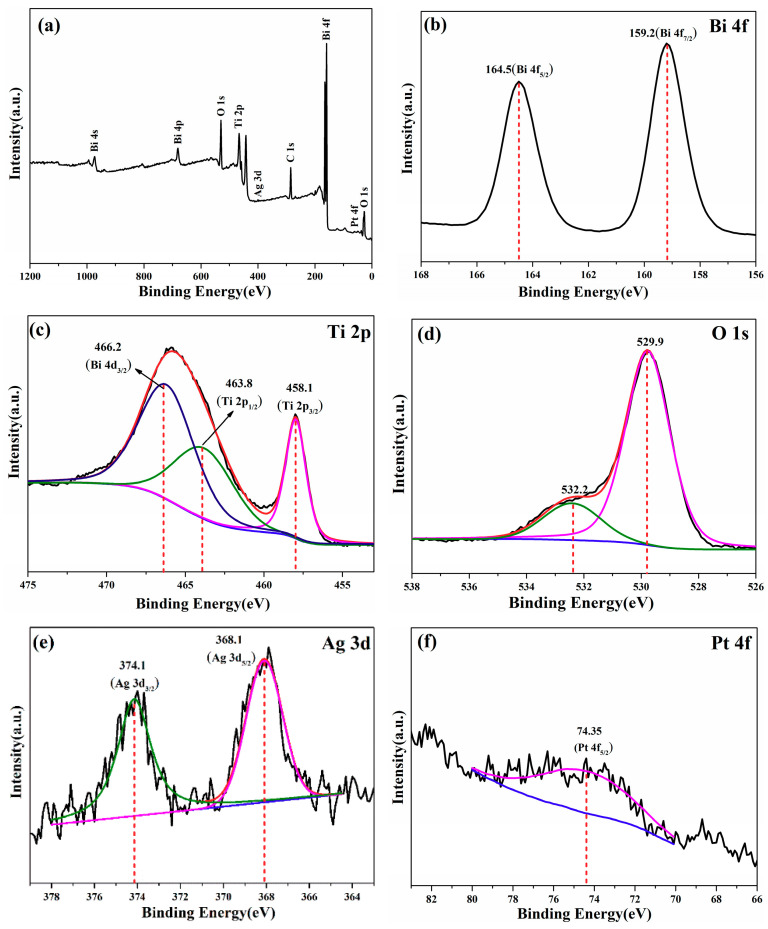
(**a**) X-ray photoelectron spectroscopy (XPS) survey scan spectrum and (**b**–**f**) high resolution XPS spectra of Bi 4f, Ti 2p, O 1s, Ag 3d and Pt 4f of (Ag_0.7_Pt_0.3_)_0.001_/Bi_4_Ti_3_O_12_.

**Figure 8 nanomaterials-10-02206-f008:**
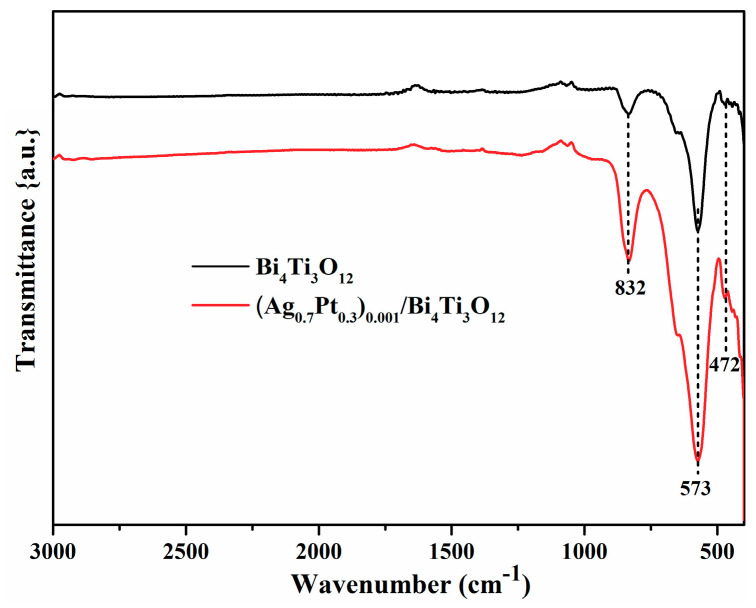
Fourier transform infrared (FTIR) spectra of Bi_4_Ti_3_O_12_ and (Ag_0.7_Pt_0.3_)_0.001_/Bi_4_Ti_3_O_12_.

**Figure 9 nanomaterials-10-02206-f009:**
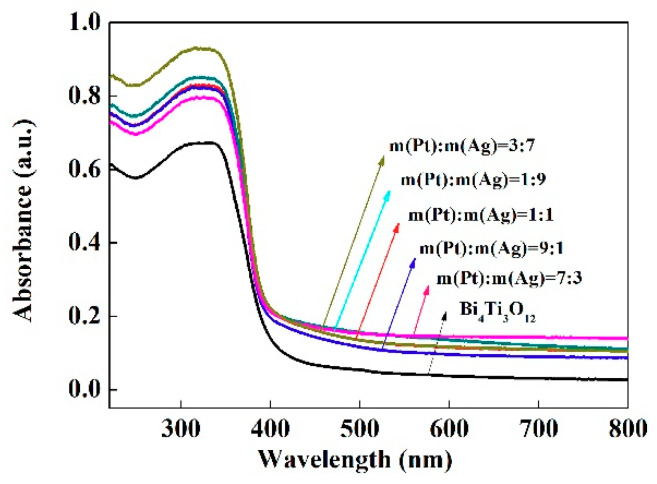
Ultraviolet-visible (UV-Vis) absorption spectra of starting Bi_4_Ti_3_O_12_ and (AgPt)_0.001_/Bi_3_Ti_4_O_12_ composite photocatalysts prepared by photoreduction method.

**Figure 10 nanomaterials-10-02206-f010:**
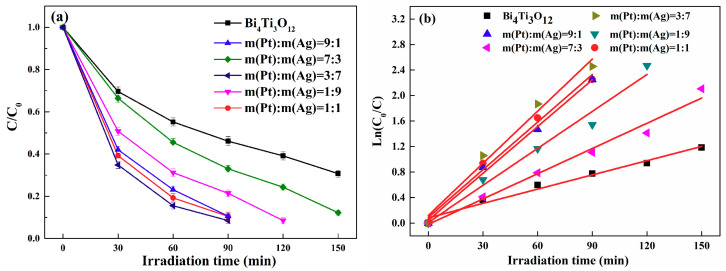
(**a**) Photocatalytic activity of pristine Bi_4_Ti_3_O_12_ and as-prepared (AgPt)_0.001_/Bi_4_Ti_3_O_12_ composite photocatalysts prepared by photocatalytic reduction method on the degradation of rhodamine B (RhB); (**b**) plot of Ln(C_0_/C) versus irradiation time for the photodegradation of RhB.

**Figure 11 nanomaterials-10-02206-f011:**
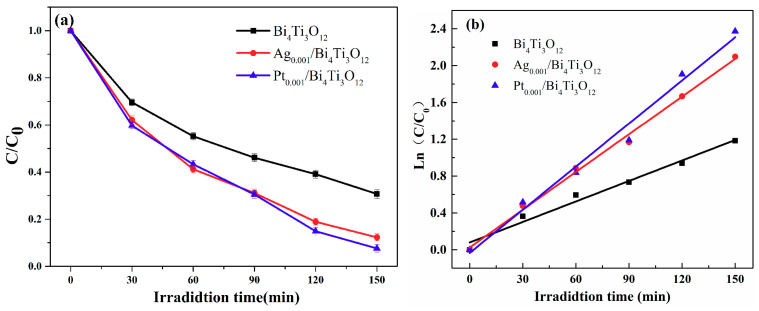
(**a**) Photocatalytic activity of pristine Bi_4_Ti_3_O_12_, Ag_0.001_/Bi_3_Ti_4_O_12_ and Pt_0.001_/Bi_3_Ti_4_O_12_ composite photocatalysts prepared by photocatalytic reduction method on the degradation of RhB; (**b**) plot of Ln(C_0_/C) versus irradiation time for the photodegradation of RhB.

**Figure 12 nanomaterials-10-02206-f012:**
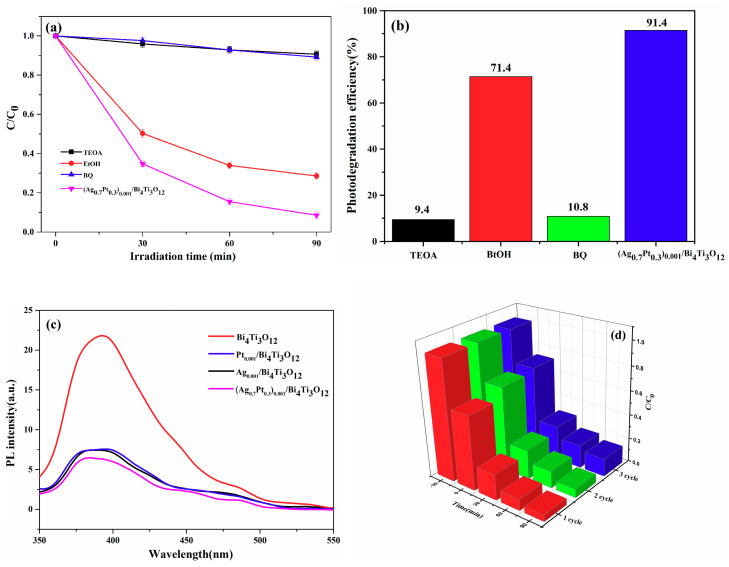
(**a**,**b**) Effects of various scavengers on the photocatalytic efficiency of (Ag_0.7_Pt_0.3_)_0.001_/Bi_4_Ti_3_O_12_; (**c**) photoluminescence spectra of Bi_4_Ti_3_O_12_ and AgPt/Bi_4_Ti_3_O_12_; (**d**) Cyclic photodegradation of RhB by (Ag_0.7_Pt_0.3_)_0.001_/Bi_4_Ti_3_O_12_ photocatalyst.

**Figure 13 nanomaterials-10-02206-f013:**
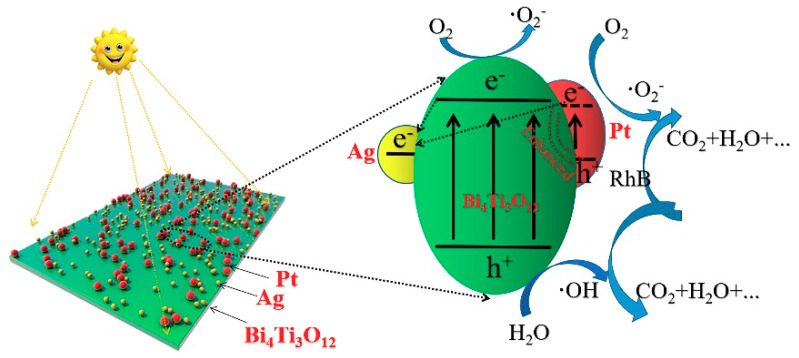
Proposed photocatalytic mechanism for degradation of RhB using as-prepared AgPt/Bi_4_Ti_3_O_12_ composite photocatalysts.

**Table 1 nanomaterials-10-02206-t001:** Synthesis conditions and batch of Bi_4_Ti_3_O_12_.

Firing Conditions	Salt Medium Composition (Molar Ratio)	Mass Ratio of Salt to Reactant
700 °C/2 h	NaCl:KCl (1:1)	1:1
800 °C/2 h		
900 °C/2 h		
700 °C/2 h	NaCl:KCl (1:1)	2:1
800 °C/2 h		
900 °C/2 h		
700 °C/2 h	NaCl:KCl (1:1)	3:1
800 °C/2 h		
900 °C/2 h		

**Table 2 nanomaterials-10-02206-t002:** Batch compositions of Ag/Bi_4_Ti_3_O_12_, Pt/Bi_4_Ti_3_O_12_ and AgPt/Bi_4_Ti_3_O_12_ photocatalysts.

Metal Composition (Mass Ratio)	Loading Capacity	Illumination Condition
Ag	0.1 wt%	300 W xenon lamp (*λ* > 400 nm) 60 min
	0.2 wt%	
	0.5 wt%	
	1.0 wt%	
	3.0 wt%	
	5.0 wt%	
Pt	0.1 wt%	
Ag:Pt (1:1)	0.1 wt%	
Ag:Pt (7:3)		
Ag:Pt (3:7)		
Ag:Pt (9:1)		
Ag:Pt (1:9)		

**Table 3 nanomaterials-10-02206-t003:** The content of Ag and Pt in the as-prepared photocatalysts.

As-Prepared Photocatalyst	Element	Content (wt%)
Ag_0.001_/Bi_4_Ti_3_O_12_	Ag	0.0347
Pt_0.001_/Bi_4_Ti_3_O_12_	Pt	0.0782
(Ag_0.7_Pt_0.3_)_0.001_/Bi_4_Ti_3_O_12_	Ag	0.0122
	Pt	0.0144

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
