# Peer review of "Preparation and Photocatalytic Performance for Degradation of Rhodamine B of AgPt/Bi4Ti3O12 Composites"

_nanomaterials, 2020, doi:10.3390/nano10112206_

Round 1

Reviewer 1 Report

This article is interesting and they report useful information but lack explanation of the novelty of this study when compared to similar type and Bi4Ti3O12/Ag3PO4 composites. Thus the authors should clarify the feasibility and significant novelty compared to the following articles(Zheng, C., Yang, H., Cui, Z. et al. A novel Bi4Ti3O12/Ag3PO4 heterojunction photocatalyst with enhanced photocatalytic performance. Nanoscale Res Lett 12, 608 (2017). https://doi.org/10.1186/s11671-017-2377-1;Highly Efficient Ag2O/Bi2O2CO3 p-n Heterojunction Photocatalysts with Improved Visible-Light Responsive Activity
Na Liang, Min Wang, Lun Jin, Shoushuang Huang, Wenlong Chen, Miao Xu, Qingquan He, Jiantao Zai, Nenghu Fang, and Xuefeng Qian
ACS Applied Materials & Interfaces 2014 6 (14), 11698-11705
DOI: 10.1021/am502481z;https://doi.org/10.1016/j.seppur.2020.116622; https://doi.org/10.1016/j.apcatb.2020.118876 etc.) which are published the same materials. Add error bars to the figure 10-12 for understanding the reproducibility of results.

Reviewer 2 Report

The manuscript entitled “Preparation and Photocatalytic Performance for Degradation of Rhodamine B of AgPt/Bi4Ti3O12 heterojunction composites” deals with the synthesis of Bi4Ti3O12powders, via a molten salt method, followed by a noble metal photoreduction method to get AgPt/Bi4Ti3O12. The characterization of AgPt/Bi4Ti3O12 was performed with XRD, SEM, TEM, XPS, FTIR and UV-vis and their ability for the photocatalytic degradation of Rhodamine B was checked. It was found that the AgPt/Bi4Ti3O12 system shows significantly improved catalytic performance.

I have some major concerns with this study:

1) Studies of the photodegradation of Rhodamine B or other model dyes are so widespread in the literature that almost no novelty can be added; It could be much more useful to study the degradation of real water contaminants;

4) The irradiation of solution with xenon lamps for photocatalysis is nowadays dated since new studies are usually carried out using solar light.

2) In addition to new data on the degradation of real water contaminants, also their photodegradation together with the photodegradation of Rhodamine B using solar light, with no catalyst, should be added;

3) The authors show the nominal chemical composition of the catalysts but no experiments have been reported to check on the real final composition of the catalysts;

4)  The proposed photocatalytic mechanism for the degradation of Rhodamine B using the AgPt/Bi4Ti3O12 composite photocatalysts is just a supposition since no real experiments are reported to state the active species.

In summary, this study is not consistent and many additional new data are necessary before its possible publication.

Round 2

Reviewer 2 Report

The manuscript was improved.